# High HER2 Intratumoral Heterogeneity Is Resistant to Anti-HER2 Neoadjuvant Chemotherapy in Early Stage and Locally Advanced HER2-Positive Breast Cancer

**DOI:** 10.3390/cancers17132126

**Published:** 2025-06-24

**Authors:** Takaaki Hatano, Tomonori Tanei, Shigeto Seno, Yoshiaki Sota, Nanae Masunaga, Chieko Mishima, Masami Tsukabe, Tetsuhiro Yoshinami, Tomohiro Miyake, Masafumi Shimoda, Kenzo Shimazu

**Affiliations:** 1Department of Breast and Endocrine Surgery, Graduate School of Medicine, Osaka University, 2-15 Yamadaoka, Suita 565-0871, Osaka, Japan; hta6878@onsurg.med.osaka-u.ac.jp (T.H.); y_sota123@onsurg.med.osaka-u.ac.jp (Y.S.); nanae.masunaga@onsurg.med.osaka-u.ac.jp (N.M.); cmishima@onsurg.med.osaka-u.ac.jp (C.M.); mtsukabe@onsurg.med.osaka-u.ac.jp (M.T.); yosinami-te@onsurg.med.osaka-u.ac.jp (T.Y.); t_miyake@onsurg.med.osaka-u.ac.jp (T.M.); mshimoda@onsurg.med.osaka-u.ac.jp (M.S.); kshimazu@onsurg.med.osaka-u.ac.jp (K.S.); 2Department of Bioinformatic Engineering, Graduate School of Information Science and Technology, Osaka University, 1-5 Yamadaoka, Suita 565-0871, Osaka, Japan; senoo@ist.osaka-u.ac.jp

**Keywords:** breast carcinoma, intratumoral heterogeneity, HER2 gene expression, neoadjuvant therapy, breast neoplasms

## Abstract

Intratumoral heterogeneity (ITH) represents a pervasive feature of breast malignancies, often undermining therapeutic efficacy. In particular, tumors with elevated ITH are prone to exhibit adaptive divergence in human epidermal growth factor receptor 2 (HER2) status when subjected to HER2-targeted interventions. In this investigation, we characterized HER2-specific ITH by examining the morphological profiles of HER2 gene amplification through fluorescence in situ hybridization (FISH) histogram distributions, derived from intratumoral HER2 copy number variation. Our study sought to clarify the prognostic relevance of extensive HER2 heterogeneity in determining tumor response to neoadjuvant anti-HER2 treatment in newly diagnosed breast cancer cases. We identified a robust correlation between marked HER2 heterogeneity and the attainment of pathological complete response (pCR) among patients with HER2-positive disease. These observations highlight HER2 ITH as a potential predictive biomarker for therapeutic efficacy. Moreover, our method of analyzing FISH distribution patterns offers a practical and translationally valuable tool for anticipating clinical outcomes in the setting of HER2-targeted adjuvant therapy.

## 1. Introduction

Intratumoral heterogeneity (ITH) refers to the presence of varying tumor cells with distinct morphological and phenotypic profiles, representing a fundamental characteristic of cancer biology [1,2,3,4]. ITH has significant implications for cancer diagnosis, therapeutic approaches, and prognosis [2,3,4,5,6,7]. ITH encompasses several aspects, including genetic mutations, epigenetic changes, and differences in cell signaling pathways [8,9]. Consequently, ITH poses substantial challenges to the development of effective cancer therapies [2,3,4,5,6,7,10].

Breast cancer tumors possess ITH, and therapeutic resistance has been observed in tumors with high-ITH [3,7,11,12]. In addition, in breast cancer tumors, HER2 expression can be heterogeneous [13]. HER2 ITH is characterized by the coexistence of at least two cellular clones with differing HER2 statuses or the presence of tumor cell subpopulations exhibiting varying HER2 protein expression levels [8].

We have reported that 18% of HER2-positive breast cancers had HER2 ITH, and other studies have indicated that 11–40% of HER2-positive breast cancers had ITH of HER2 expression within the tumor, which may cause inaccurate HER2 status assessments, potentially affecting therapeutic decision-making [14,15,16,17,18,19,20]. Therefore, accurate HER2 ITH assessment is critical for optimizing therapeutic approaches for HER2-positive breast cancers. Although HER2-positive breast cancer is more aggressive than other subtypes, it demonstrates promising responses to HER2-targeted therapies, including trastuzumab and pertuzumab [11,21,22,23]. Advances in HER2-targeted therapies have significantly improved the prognosis of patients with HER2-positive breast cancer [24,25,26].

HER2 evaluation mainly aims to predict the responsiveness to anti-HER2 therapies. HER2 status is determined using fluorescence in situ hybridization (FISH) to evaluate HER2 gene expression or immunohistochemistry (IHC) to assess HER2 protein expression in primary breast cancer tissues [27,28]. HER2 ITH may compromise the accuracy of HER2 status assessment and influence therapeutic decision-making, necessitating the development of technologies to identify HER2 ITH in clinical settings and the establishment of effective HER2-targeted therapies for tumors with heterogeneous HER2 expression. HER2 ITH is primarily evaluated using the following three methods: HER2 IHC, HER2 dual-color ISH, and HER2 gene–protein assay (HER2 GPA), which combines HER2 IHC and dual-color ISH [27,28,29]. Studies have demonstrated that HER2 ITH, as detected using HER2 GPA, is associated with therapeutic resistance to anti-HER2 neoadjuvant chemotherapy (NAC) in patients with HER2-positive breast cancer [30]. However, these methods depend on visual assessment by pathologists, which introduces interobserver variability and subjective interpretation, making them impractical for routine application.

To address these limitations, we have developed a novel approach for evaluating HER2 ITH in a simple, objective, and automated manner [14]. This method employs the Gaussian mixture model (GMM) analysis to objectively and automatically evaluate HER2 ITH on the basis of FISH histograms. This approach enables the straightforward and objective assessment of HER2 ITH to be analyzed on the basis of the distribution of HER2 gene copy number histograms. In our recent study, this method demonstrated efficacy in assessing HER2 ITH and predicting prognosis, including progression-free survival (PFS) and overall survival (OS), in patients with HER2-positive breast cancer [14]. Conversely, Seol et al. reported that HER2 heterogeneity was associated with reduced disease-free survival (DFS), with a DFS rate of only 25% in patients receiving adjuvant trastuzumab therapy. The association between HER2 ITH and clinical outcomes remains inconsistent across studies.

We here aimed to elucidate the relationship between HER2 ITH and resistance to anti-HER2 therapies by leveraging a novel approach utilizing HER2 gene copy number histograms. We investigated the correlation between HER2 ITH and anti-HER2 NAC response, focusing on the pathological complete response (pCR) rates in HER2-positive breast cancer.

## 2. Materials and Methods

### 2.1. HER2 FISH Assay

HER2 FISH assays were conducted on all tumor tissues of VAB before NAC. HER2 gene signals and their ratios were assessed using PathVysion HER2 DNA Probe Kits (SRL Inc., Tokyo, Japan). FFPE tumor tissue slides were dried, and 20 µL of DAPI counterstain solution was applied. The number of orange-fluorescent HER2/neu signals and green-fluorescent CEP17 signals was counted in more than 20 cell nuclei using a fluorescence microscope. The HER2 FISH assay was performed using either manual or semiautomated protocols by SRL Inc. The ratio of HER2/neu to CEP17 signal counts was determined, and tumors were classified as HER2-positive when more than two HER2 gene copies per cell were detected. Experienced molecular pathologists assessed the HER2 status in accordance with the updated 2018 ASCO/CAP HER2 guidelines. The details have been previously described [14].

### 2.2. HER2 ITH Assessment

We evaluated HER2 ITH using the same approach as previously reported [14]. In brief, we obtained scanned images from FISH diagnostic reports and extracted the HER2 FISH signal count distribution by applying optical character recognition (OCR) and other image processing techniques to reconstruct the FISH signal histograms. Subsequently, we modeled these distributions using the GMM. Specifically, the HER2 gene copy number distribution within each tumor sample was assumed to be represented by two normal components. The mixture model was defined as follows:(*x*) = *π*1*N*(*x*│*μ*1, *σ*1) + *π*2*N*(*x*│*μ*2, *σ*2),
where *N* denotes a normal distribution, and *π*i (with means *μ*i and standard deviations *σ*i) refers to the mixture weights for each component. From the estimated parameters, we derived an ITH measure that captures the presence of a subpopulation with HER2 copy numbers of <2 while the overall average copy number remains ≥2.

HER2 ITH was evaluated based on the patterns of HER2 FISH signal distributions. Tumors were classified into two distinct groups according to their distribution patterns as illustrated in Figure 1: a double-peaked distribution indicative of high heterogeneity (HH), and a single-peaked distribution reflective of low heterogeneity (LH). The pathological complete response (pCR) rates after neoadjuvant chemotherapy (NAC) were subsequently compared between these two groups. Diagnostic images were analyzed using Python (v3.11.4) along with the libraries “pillow” (v9.4.0), “pyOCR” (v0.8.5), and “pytesseract” (v0.3.13).

### 2.3. Statistical Analyses

All patients were stratified into high heterogeneity (HH) and low heterogeneity (LH) cohorts, and comparative analyses were undertaken based on a range of clinicopathological characteristics. The chi-square test was employed for evaluating the association between HER2 ITH and clinicopathological parameters. The analysis of HER2 ITH and the therapeutic efficacy of preoperative chemotherapy with anti-HER2 agents was performed using Fisher’s exact test or the chi-square test to examine the relationship between the pCR and the incomplete response. Cox proportional hazard models were employed for the univariate and multivariate analyses of various pathological and biological factors associated with pathological outcomes. All statistical analyses were performed using R software (v.4.2.3; 14), with *p* < 0.05 for two-sided tests considered statistically significant, and the results were reported.

### 2.4. Data Availability

This study was conducted with the approval of the Medical Ethics Committee of Osaka University (approval ID: 22080 [T1]). Employing an opt-out consent framework, participants were considered enrolled unless they proactively chose not to participate, in alignment with institutional guidelines for informed consent in observational research. Notification and detailed study information were made publicly accessible via the institutional website, and individuals who opted out were systematically excluded from data collection and analysis.

## 3. Results

### 3.1. Association Between HER2 ITH and Patient Characteristics

Illustrative examples of HER2 FISH signal distribution histograms are presented in Figure 1. Based on the graphical interpretation of HER2 amplification patterns—categorized as either biphasic or monophasic—HER2-positive breast carcinomas were stratified into high heterogeneity (HH) and low heterogeneity (LH) cohorts. Among the 97 HER2-positive cases analyzed, 18 (18.6%) were assigned to the HH group, while 79 (81.4%) were designated as LH. The average HER2 FISH ratio in the HH group was 2.5, whereas a markedly higher mean ratio of 5.3 was observed in the LH group. Notably, no statistically significant difference was detected in the distribution of HER2 immunohistochemistry (IHC) scores (3+ or 2+) between the two groups, as detailed in Table 1. Moreover, the HH group exhibited a tendency toward being estrogen receptor (ER)-positive (*p* = 0.12), premenopausal (*p* = 0.11), positive for pretreatment lymph node (LN) metastasis (*p* = 0.10), and positive for resected LN metastasis (*p* = 0.12); however, no statistically significant associations were noted. The proportion of HH of HER2-positive tumors following the ER and progesterone receptor (PgR) status is depicted in Appendix A. The ER+ and PgR+ tumors exhibited the highest HH frequency (ER+ and PgR+, 26.9% [7/26]; ER+ and PgR−, 23.1% [6/26]; ER− and PgR−, 11.6% [5/43]; and ER− and PgR+: 0% [0/2]).

The intratumoral heterogeneity (ITH) of HER2 using a fluorescence microscope imaging (a) is identified on the basis of the distribution patterns of HER2 FISH amplification histograms with the HER2 gene copy number within the tumor samples. The representative right histograms (b) are classified into two distinct groups as follows: the biphasic graph (HH group) (A) and the monophasic graph (LH group) (B). The representative images on the left indicate HER2 FISH imaging in the tumor cell nuclei, showing HER2 (red signals) and CEP17 (green signals), with DAPI (blue) staining the nuclei.

### 3.2. HER2 ITH and Patient Prognosis

The pCR rate to anti-HER2 NAC in patients with HER2-positive breast cancer in the HH group was 28% (5/18 cases), which was significantly lower than that in the LH group at 65% (51/79 cases), indicating that tumors in the HH group exhibited a markedly reduced response to HER2-targeted neoadjuvant therapy and resistance to anti-HER2 agents (univariate analysis, *p* < 0.01) (Figure 2). Univariate analysis further demonstrated that ER negativity, PgR negativity, and high histological grade (HG3) were significantly associated with the pCR rate (pCR rate: ER, *p* = 0.0003; PgR, *p* = 0.001; and HG3, *p* = 0.09) (Table 2).

Multivariate analysis of the pathological factors revealed that ER status and HER2 ITH were significantly correlated with the pCR rate (multivariate analysis: ER, *p* = 0.03; HER2 heterogeneity, *p* = 0.02) (Table 2).

The pCR rate to anti-HER2 NAC (B) comparing the HH group with the LH group (pCR, *p* < 0.01 HH group: 28% [5/18] vs. LH group: 65% [51/79]).

## 4. Discussion

This study aimed to elucidate the correlation between HER2 ITH, assessed using histograms with HER2 gene copy number by FISH, and resistance of anti-HER2 NAC in HER2-positive breast cancer patients (Appendix A). The HH group had a prevalence of 18.6%, consistent with our previous study [11]. The HH group demonstrated significantly lower HER2 FISH ratios and higher ER positivity rates than the LH group. Luminal HER2-positive tumors with ER and PgR positivity exhibited the highest prevalence of HH tumors (26.9%, Appendix A). These findings support the idea that ITH is fueled by the diverse biological characteristics of cancer cells.

The pCR following anti-HER2 NAC is considered a surrogate marker for long-term survival in HER2-positive breast cancer. In our study, the overall pCR rate was 59%, which is comparable to the approximately 50% pCR rate reported in large clinical trials [18]. Notably, using histograms with HER2 gene copy number, the HH group demonstrated a markedly lower pCR rate (28%) than the LH group (65%), revealing that the HH group is significantly associated with reduced anti-HER2 NAC responsiveness (Table 2).

HER2-positive breast cancers with ITH may demonstrate anti-HER2 therapy resistance due to distinct biological characteristics. Hou et al. and Horii et al. reported that the HER2 GPA method combining HER2 IHC and dual-color ISH showed lower pCR rates in the HH group than those in the LH group, identifying HER2 ITH as a predictor of incomplete response [25,26]. However, the HER2 GPA method does not analyze the distribution of HER2 gene copy numbers within tumor samples and relies on pathologists’ visual assessments, resulting in significant interobserver variability and subjective interpretation. The quantitative evaluation of HER2 gene copy numbers within tumor samples, facilitating a clearer HER2 ITH analysis, constitutes a key strength of our study. Miglietta et al. highlighted significant interobserver variability in HER2 IHC evaluations, advocating for alternative approaches, including molecular testing and digital pathology [27]. In our study, hormone receptor (HR)-negative tumors demonstrated higher pCR rates with anti-HER2 NAC, consistent with studies identifying HR negativity and elevated Ki-67 proliferation indices as potential predictors of high pCR rates [28]. Our study noted no predictive value for HER2 IHC scores regarding pCR (HER2 IHC 3+ [54%] vs. 2+ [43%]; *p* = 0.4).

Recent breakthroughs in digital image analysis have highlighted its promise in rapidly and objectively evaluating HER2 IHC and the effectiveness of anti-HER2 therapies. However, the digital analysis of HER2 IHC heterogeneity remains underdeveloped. Although our study did not compare HER2 IHC with intratumoral HER2 gene copy numbers, our HER2 ITH method using FISH histograms demonstrated a strong potential for a more accurate prediction of therapeutic outcomes. This strategy provides a straightforward cost-effective method that merely involves interpreting the FISH diagnostic report histograms and performing image analysis. This study has several limitations. First, the sample size is relatively small, which may limit the generalizability of our findings. A larger, multicenter cohort study is necessary to validate our results. Second, our study is retrospective in nature, which may introduce selection bias. Prospective studies are required to confirm the clinical utility of our approach. Third, while we assessed key biomarkers and treatment responses, other potential confounding factors—such as tumor heterogeneity, genetic variations, molecular aberrations, and alterations in signaling pathways—were not comprehensively analyzed. Previous studies have suggested that neoadjuvant chemotherapy (NAC) may induce changes in intracellular signaling pathways [31], and further investigation may facilitate the development of more effective targeted therapies. Fourth, this study was limited to formalin-fixed, paraffin-embedded (FFPE) specimens. It is essential to validate whether our findings are reproducible using fresh breast cancer tissues and HER2-amplified breast cancer models in vivo. Fifth, treatment resistance may not be solely attributable to intratumoral heterogeneity. Alternative mechanisms may contribute to tumor progression, such as insufficient upregulation of HER2 protein expression. Quantitative proteomic approaches assessing total HER2 receptor levels may provide greater insight into these possibilities [32,33,34]. We intend to pursue further in-depth studies, incorporating comprehensive molecular profiling to refine and validate our findings.

Despite the limitations of the small cohort size and the missing Ki-67 proliferation index data, our study successfully detected statistically significant differences in HER2 ITH between patients with and without pCR (Graphic Abstract). This innovative analytical approach could establish a novel classification system for cancer heterogeneity, potentially enhancing predictions of therapeutic efficacy in HER2-positive breast cancer and demonstrating clinical applicability.

Furthermore, this analysis holds promise for ITH evaluation in HER2-low breast cancer, which frequently represents a more heterogeneous population than HER2-positive breast cancer. Modi et al. reported that treatment with trastuzumab deruxtecan significantly improves the PFS and OS compared with chemotherapy in patients with HER2-low metastatic breast cancer [35]. These findings suggest that HER2-low breast cancers represent a heterogeneous population with differing prognoses and sensitivities to systemic treatments. Furthermore, our method may facilitate precise ITH evaluation for predicting therapeutic responses in HER2-low breast cancers. We intend to utilize this analytical method for evaluating ITH in HER2-low metastatic recurrent breast cancer and investigating the therapeutic efficacy of trastuzumab deruxtecan.

Here, HER2 ITH was identified as a predictive marker of anti-HER2 NAC resistance in HER2-positive breast cancer. The HER2 FISH distribution analysis presented in this study is a straightforward, cost-effective, and distinct method with potential clinical utility in predicting anti-HER2 NAC resistance. In the future, this analysis may hold potential for evaluating ITH and predicting therapeutic efficacy in HER2-low breast cancers.

## 5. Conclusions

HER2 ITH assessment may provide valuable insights into predicting treatment outcomes in HER2-positive breast cancer. Our novel approach of the HER2 ITH method using FISH histograms could be a useful tool for predicting anti-HER2 NAC resistance.

## Figures and Tables

**Figure 1 cancers-17-02126-f001:**
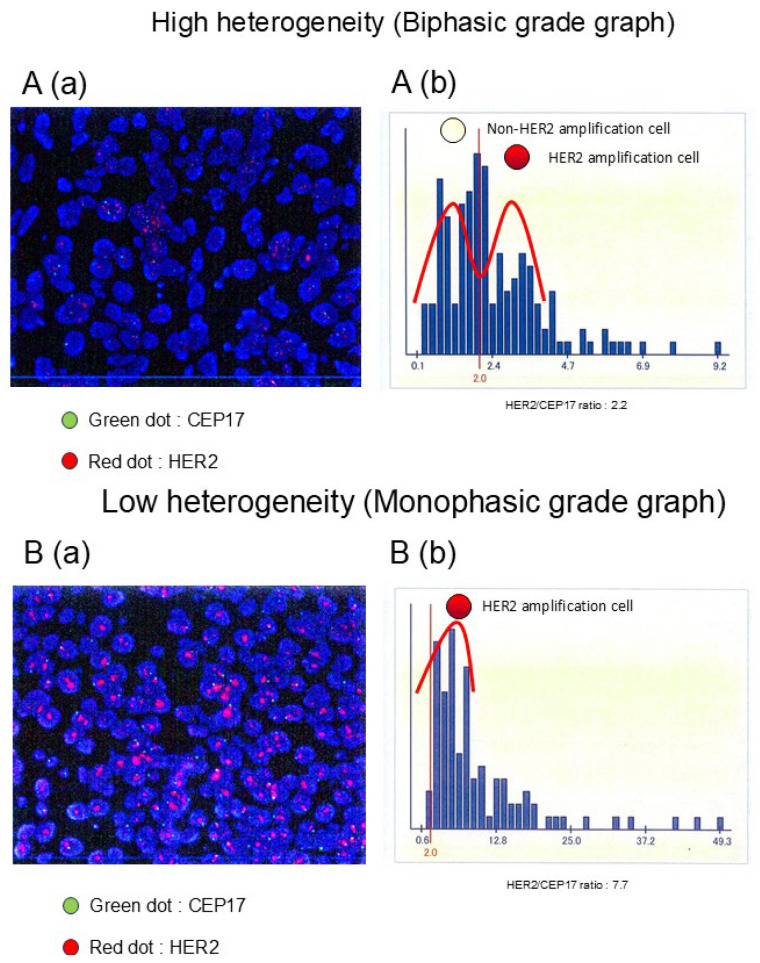
Representative histograms of the HER2 FISH distributions. HER2/CEP17 signal (a), histogram analysis of HER2 FISH signals (b), classifying the cases into the high heterogeneity (**A**) and low heterogeneity groups (**B**).

**Figure 2 cancers-17-02126-f002:**
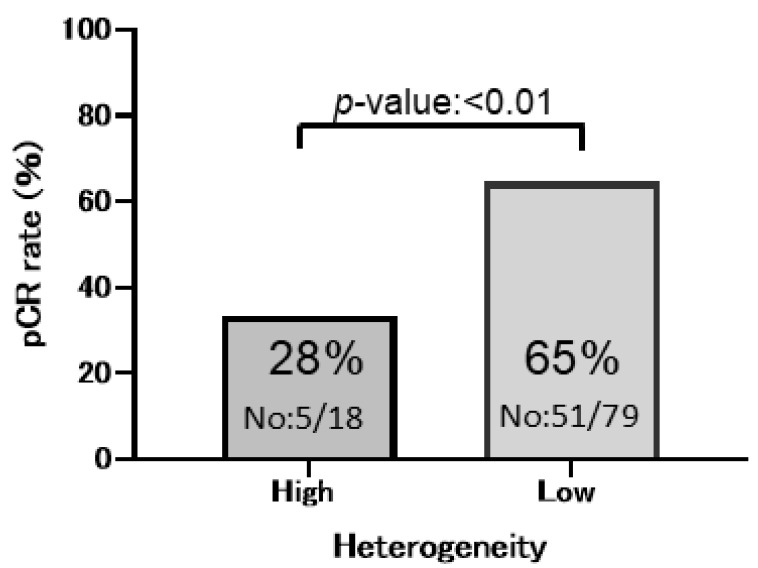
Pathological complete response (pCR) rate to anti-HER2 neoadjuvant chemotherapy (NAC) in patients with HER2-positive breast cancer in both groups.

**Table 1 cancers-17-02126-t001:** Association between patient characteristics and HER2 heterogeneity.

		HER2 Heterogeneity		
	Total	High	Low	*p* Value
	n = 97%	n = 18%	n = 79%	
Menopause				
pre	47 (48.5%)	12 (66.7%)	35 (44.3%)	0.12
post	50 (51.5%)	6 (33.3%)	44 (55.7%)	
Histological type				
IDC	95 (97.9%)	17 (94.4%)	78 (98.8%)	0.34
ILC	2 (2.1%)	1 (5.6%)	1 (1.3%)	
Histological grade				
1	5 (5.2%)	0 (0.0%)	5 (6.3%)	0.25
2	33 (34.0%)	9 (50.0%)	24 (30.4%)	
3	59 (60.8%)	9 (50.0%)	50 (63.3%)	
Tumor size				
T1	7 (7.2%)	1 (5.6%)	6 (7.6%)	1.00
T2–T4	90 (92.8%)	17 (94.4%)	73 (92.4%)	
Pretreatment LN metastasis				
Negative	34 (35.1%)	3 (16.7%)	31 (39.2%)	0.10
Positive	63 (64.9%)	15 (83.3%)	48 (60.8%)	
ER				
Positive	52 (53.6%)	13 (72.2%)	40 (50.6%)	0.12
Negative	45 (46.4%)	5 (27.8%)	39 (49.4%)	
PgR				
Positive	28 (28.9%)	7 (38.9%)	21 (26.6%)	0.39
Negative	69 (71.1%)	11 (61.1%)	58 (73.4%)	
HER2 IHC				
3+	90 (92.8%)	16 (88.9%)	74 (93.7%)	0.61
2+	7 (7.2%)	2 (11.1%)	5 (6.3%)	
HER2 ratio				
Mean	4.8 ± 2.2	2.5 ± 0.5	5.3 ± 2.2	1.23 × 10^−17^
Stage				
I	4 (5.1%)	0 (0.0%)	4 (5.1%)	1.00
II	74 (76.3%)	14 (77.8%)	60 (75.9%)	
III	19 (19.6%)	4 (22.2%)	15 (19.0%)	

**Table 2 cancers-17-02126-t002:** Univariate and multivariate analyses of pathological factors on the pCR rate to anti-HER2 NAC in HER2-positive patients.

				Univariate Analysis			Multivariate Analysis		
Characteristic	pCR Rate(%)	*N*	pCR(N)	OR	95% CI	*p* Value	OR	95% CI	*p* Value
Total	58	97	56						
ER									
Positive	40	52	21						
Negative	78	45	35	0.194	0.0791–0.474	0.0003	0.321	0.1130–0.912	0.03
PgR									
Positive	32	28	9						
Negative	68	69	47	0.222	0.0865–0.568	0.001	0.38	0.1230–1.170	0.09
Histological grade									
1 or 2	45	38	17						
3	66	59	39	1.79	0.8970–3.59	0.09	1.84	0.8560–3.970	0.11
HER2 heterogeneity									
Low	65	79	51						
High	28	18	5	0.211	0.0682–0.654	0.006	0.241	0.0699–0.828	0.02

## Data Availability

The data presented in this study are available on request from the corresponding author (accurately indicate status).

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
