# Peer review of "High HER2 Intratumoral Heterogeneity Is Resistant to Anti-HER2 Neoadjuvant Chemotherapy in Early Stage and Locally Advanced HER2-Positive Breast Cancer"

_cancers, 2025, doi:10.3390/cancers17132126_

Round 1

Reviewer 1 Report

Comments and Suggestions for Authors

This study evolves the data that largely presents the correlation between ITH and cPR.  Yet, it suffers from a lack of molecular insights that explain why high ITH leads to Her-2 Neoadjuvant Chemo resistant.  Are there any molecular aberrations or signal pathway alterations responsible for the treatment resistance?  Uncovering the molecular perturbation can aid the developing targeted therapy, by correcting the molecular aberrations.

This study solely built on the FFPE specimens, in correlations with clinical outcome cPR. Can this finding be substantiated in fresh BC samples or in animal models engineered with Her-2 amplification-induced BC?  

Minor issues:

Abbreviation as being “growth factor receptor 2 (HER2)” shall not be repeated in line 60, once it is established in the prior section.

Layout in Table 1 with 2 vertical subgroups seems to be rather confusing and it results in difficulties of pleasing readers’ eyes.  Furthermore, data from Table 1 seems to be dispensable.  Perhaps, Table 1 shall be considered to be removed from the main text body or moved to Supplemental data.

Legends for Figures and Tables are excluded from the manuscript that shall be considered to be added.

What does HG mean, in the heading of Table 3?

Abbreviations (lines 309 ~ 326) shall be listed in an alphabetic order.

Grammatic concern: our approach to analyzing FISH (line 24)

Author Response

Comment1:

This study evolves the data that largely presents the correlation between ITH and cPR. Yet, it suffers from a lack of molecular insights that explain why high ITH leads to Her-2 Neoadjuvant Chemo resistant. Are there any molecular aberrations or signal pathway alterations responsible for the treatment resistance? Uncovering the molecular perturbation can aid the developing targeted therapy, by correcting the molecular aberrations.

Response1:

Previous reports have suggested that NAC treatment may cause changes in signal pathways.

Further analysis may contribute to the development of targeted therapy. But this study was a retrospective study that used only clinical data. This is an important topic, I have added it to Discussion section. (Added in the discussion, page10, (lines 267~ 273))

Comment2:

This study solely built on the FFPE specimens, in correlations with clinical outcome cPR. Can this finding be substantiated in fresh BC samples or in animal models engineered with Her-2 amplification-induced BC? 

Response2:

It is necessary to confirm whether the results can be reproduced using fresh breast cancer tissue samples and animal models engineered with HER2 amplification induced breast cancer.

We plan to conduct more detailed research. (Added in the discussion, page10, (lines 273~ 276))

Minor issues:

Comment3:

Abbreviation as being “growth factor receptor 2 (HER2)” shall not be repeated in line 60, once it is established in the prior section.

Response3:

I fixed it according to your advice. (Fixed in page2, (lines 60))

Comment4:

Layout in Table 1 with 2 vertical subgroups seems to be rather confusing and it results in difficulties of pleasing readers’ eyes. Furthermore, data from Table 1 seems to be dispensable. Perhaps, Table 1 shall be considered to be removed from the main text body or moved to Supplemental data.

Response4:

Following your advice, it was moved to Supplementary materials. (Moved from page3 to supplementary table S1)

Comment5:

Legends for Figures and Tables are excluded from the manuscript that shall be considered to be added.

Response5:

We added Legends to Figures. (Fixed in page4, (Figure 1))

Comment6:

What does HG mean, in the heading of Table 3?

Response6:

HG means histological grade. The HG notation has been made more detailed. (Fixed in page9, (Table 2))

Comment7:

Abbreviations (lines 309 ~ 326) shall be listed in an alphabetic order.

Response7:

We rearranged abbreviations in alphabetical order. (Fixed in page12, (lines 341~ 370))

Comment8:

Grammatic concern: our approach to analyzing FISH (line 24)

Response8:

We changed it to a more grammatically correct expression. (Fixed in page2, (lines 60~ 61))

Reviewer 2 Report

Comments and Suggestions for Authors

The manuscript is well written and discusses a quantitative method to address the ITH of HER2+ breast cancers by analyzing FISH data. This is an interesting study and has clinical significance if the analytical technique could be translatable to a clinical setting.

How can we analyze ITH in HER2-1+ or low samples that have the potential to undergo clonal selection and generate a HER2-high population of cells post-NAC therapy?

How was the GMM analysis performed? Was any particular software used for the purpose?

The reference 10 cited on page 4 of 12, line 119, and reference 11 cited in line 117 do not correspond to the methodology claimed to be published previously. Both these references are review papers.  It would be ideal to have the exact details cited correctly.

The supplementary figure S1 lacks clarity even when magnified and should be enhanced following the journal guidelines.

Author Response

Comment1:

The manuscript is well written and discusses a quantitative method to address the ITH of HER2+ breast cancers by analyzing FISH data. This is an interesting study and has clinical significance if the analytical technique could be translatable to a clinical setting.

How can we analyze ITH in HER2-1+ or low samples that have the potential to undergo clonal selection and generate a HER2-high population of cells post-NAC therapy?

Response1:

According to Japanese guidelines, NAC is only performed on HER2-positive breast cancer, and not on those with low HER2 expression. So, this study did not include low HER2 expression. The relationship between trastuzumab deruxtucan and heterogeneity in cases of metastatic or recurrent breast cancer with low HER2 expression is a topic for future research.

Comment2:

How was the GMM analysis performed? Was any particular software used for the purpose?

Response2:

We did not use any off-the-shelf software packages. Assuming a one-dimensional, two-component Gaussian mixture model, we implemented an algorithm in Python from scratch to estimate its parameters.

Comment3:

The reference 10 cited on page 4 of 12, line 119, and reference 11 cited in line 117 do not correspond to the methodology claimed to be published previously. Both these references are review papers.  It would be ideal to have the exact details cited correctly.

Response3:

The cited articles has been corrected. (Fixed in page4, (lines 119 and 121))

Comment4:

The supplementary figure S1 lacks clarity even when magnified and should be enhanced following the journal guidelines.

Response4:

The image used was unclear, so it will be reinserted with improved accuracy. (Fixed in supplementary figure S1)

Reviewer 3 Report

Comments and Suggestions for Authors

Whether the HER2 amplification in breast cancer is the cause of the cancer or merely an accelerator of the cancer in the single patient can not be clarified at the moment

The authors show however without doubt that low and unevenly distributed HER 2 gene numbers as measured by FISH in the tissue correlate with the clinical effect of targeted treatment against the HER2 protein receptor

But it has not been shown that it is the heterogeneity and not the number of HER 2 receptors per cell that is the cause for the treatment failure

Though there is a strong correlation between gene number and HER 2 tissue concentration in general, this may not be true for the single patient. There may be other mechanisms driving the cancer in these patients - or the production of HER2 growth receptors has not yet reached its full level. So the cause of the treatment failure could actually be a lower number of receptors per cell and not the heterogeneity. A logical reason for Herceptin failure

If not possible for the authors at least this possibility should be discussed

To prove this the authors will need to measure total level of HER 2 receptors in the tissue, per mg tissue or preferably per cell by quantitative preoteome technologies

Author Response

Comment1:

Whether the HER2 amplification in breast cancer is the cause of the cancer or merely an accelerator of the cancer in the single patient can not be clarified at the moment

The authors show however without doubt that low and unevenly distributed HER 2 gene numbers as measured by FISH in the tissue correlate with the clinical effect of targeted treatment against the HER2 protein receptor

But it has not been shown that it is the heterogeneity and not the number of HER 2 receptors per cell that is the cause for the treatment failure

Though there is a strong correlation between gene number and HER 2 tissue concentration in general, this may not be true for the single patient. There may be other mechanisms driving the cancer in these patients - or the production of HER2 growth receptors has not yet reached its full level. So the cause of the treatment failure could actually be a lower number of receptors per cell and not the heterogeneity. A logical reason for Herceptin failure

If not possible for the authors at least this possibility should be discussed

To prove this the authors will need to measure total level of HER 2 receptors in the tissue, per tissue or preferably per cell by quantitative preoteome technologies

Response1:

As you point out, heterogeneity is not necessarily the cause of treatment resistance. The lower number of receptors per cell gene may lead to resistance to treatment. It is also important to confirm that the technique quantitatively measures HER2 receptors. But this study was a retrospective study that used only clinical data. This is an important thing, I have added it to Discussion section. (Added in the discussion, page10, (lines 276~ 281))

Reviewer 4 Report

Comments and Suggestions for Authors

The manuscript by Hatano et al. explores the relationship between HER2 intratumoral heterogeneity and resistance to anti-HER2 neoadjuvant chemotherapy in early and locally advanced HER2-positive breast cancer. Overall, the study is well-designed and addresses a specific research question with insightful, albeit somewhat limited, results.

Lines 128–130: The authors define HER2 copy number thresholds for high and low as >2 and <2, respectively. This seems unusual, as the lower threshold would more appropriately be defined as ≤2.

Author Response

Commnet1:

The manuscript by Hatano et al. explores the relationship between HER2 intratumoral heterogeneity and resistance to anti-HER2 neoadjuvant chemotherapy in early and locally advanced HER2-positive breast cancer. Overall, the study is well-designed and addresses a specific research question with insightful, albeit somewhat limited, results.

Lines 128–130: The authors define HER2 copy number thresholds for high and low as >2 and <2, respectively. This seems unusual, as the lower threshold would more appropriately be defined as ≤2.

Response1:

Thank you for your checking our manuscript and pointing out the mistakes.As you pointed out, FISH ratio = 2 was missing, so we added it. (Fixed in page5, (lines 132))

The definition of HER2 positivity has been revised in accordance with the guidelines.

Round 2

Reviewer 1 Report

Comments and Suggestions for Authors

This manuscript can be accepted for publication.

Comments on the Quality of English Language

English Language can be improved by Editorial Officers